# An Update on the Study of the Molecular Mechanisms Involved in Autophagy during Bacterial Pathogenesis

**DOI:** 10.3390/biomedicines12081757

**Published:** 2024-08-05

**Authors:** Md Ataur Rahman, Amily Sarker, Mohammed Ayaz, Ananya Rahman Shatabdy, Nabila Haque, Maroua Jalouli, MD. Hasanur Rahman, Taslin Jahan Mou, Shuvra Kanti Dey, Ehsanul Hoque Apu, Muhammad Sohail Zafar, Md. Anowar Khasru Parvez

**Affiliations:** 1Department of Neurology, University of Michigan, Ann Arbor, MI 48109, USA; 2Global Biotechnology & Biomedical Research Network (GBBRN), Department of Biotechnology and Genetic Engineering, Faculty of Biological Sciences, Islamic University, Kushtia 7003, Bangladesh; 3Department of Microbiology, Jahangirnagar University, Savar 1342, Bangladesh; amilysarkerjumi@gmail.com (A.S.); mohammed.ayaz.ju@gmail.com (M.A.); ananyarahman1234@gmail.com (A.R.S.); 45289nabila@juniv.edu (N.H.); moumicro@juniv.edu (T.J.M.); shuvra.dey@juniv.edu (S.K.D.); 4Department of Biology, College of Science, Imam Mohammad Ibn Saud Islamic University (IMSIU), Riyadh 11623, Saudi Arabia; mejalouli@imamu.edu.sa; 5Department of Biotechnology and Genetic Engineering, Faculty of Life Sciences, Bangabandhu Sheikh Mujibur Rahman Science and Technology University, Gopalganj 8100, Bangladesh; hasanurrahman.bge@gmail.com; 6Department of Biomedical Science, College of Dental Medicine, Lincoln Memorial University, Knoxville, TN 37923, USA; ehoquea@umich.edu; 7Department of Internal Medicine, Division of Hematology and Oncology, University of Michigan, Ann Arbor, MI 48109, USA; 8Department of Restorative Dentistry, College of Dentistry, Taibah University, Al Madinah 41311, Saudi Arabia; drsohail_78@hotmail.com; 9School of Dentistry, University of Jordan, Amman 11942, Jordan; 10Department of Dental Materials, Islamic International Dental College, Riphah International University, Islamabad 44000, Pakistan

**Keywords:** autophagy, xenophagy, bacteria, pathogenesis, bacterial toxins

## Abstract

Autophagy is a unique catabolic process that degrades irrelevant or damaged components in eukaryotic cells to maintain homeostasis and eliminate infections from pathogenesis. Pathogenic bacteria have developed many autophagy manipulation techniques that affect host immune responses and intracellular bacterial pathogens have evolved to avoid xenophagy. However, reducing its effectiveness as an innate immune response has not yet been elucidated. Bacterial pathogens cause autophagy in infected cells as a cell-autonomous defense mechanism to eliminate the pathogen. However, harmful bacteria have learned to control autophagy and defeat host defenses. Intracellular bacteria can stimulate and control autophagy, while others inhibit it to prevent xenophagy and lysosomal breakdown. This review evaluates the putative functions for xenophagy in regulating bacterial infection, emphasizing that successful pathogens have evolved strategies to disrupt or exploit this defense, reducing its efficiency in innate immunity. Instead, animal models show that autophagy-associated proteins influence bacterial pathogenicity outside of xenophagy. We also examine the consequences of the complex interaction between autophagy and bacterial pathogens in light of current efforts to modify autophagy and develop host-directed therapeutics to fight bacterial infections. Therefore, effective pathogens have evolved to subvert or exploit xenophagy, although autophagy-associated proteins can influence bacterial pathogenicity outside of xenophagy. Finally, this review implies how the complex interaction between autophagy and bacterial pathogens affects host-directed therapy for bacterial pathogenesis.

## 1. Introduction

The autophagy defense mechanism is an inherent immunological response to bacterial infection. When a bacterial infection occurs, various host factors and signaling pathways trigger autophagy, which causes pathogens to be encapsulated in double-membrane structures called autophagosomes [1] and dispatches intracellular microorganisms to lysosomes for destruction [2]. The autophagy machinery targets pathogens through direct protein–protein interactions between selective autophagy receptors and bacterial effector proteins and ubiquitin-dependent interactions with and recognition of ubiquitinated cargo [3]. Inhibition of autophagy refers to the process by which certain bacteria can halt autophagy to some extent by interfering with the essential components of the autophagy mechanism [4]. Through the interaction of bacterial effector proteins with autophagy components, bacteria can avoid being killed by autophagy, and certain strains can even exploit autophagy to promote the infection of host cells [5]. Autophagy proteins are engaged in numerous processes besides degrading cellular proteins, some crucial during bacterial infections [6]. The antimicrobial mechanism of autophagy in several bacterial species and the induction and recognition processes have been clarified; nevertheless, the exact mechanism is not entirely understood.

In the present study, autophagy is an intrinsic immunological mechanism in response to bacterial infection. Bacteria have been recognized as subjects of selective autophagy; a phenomenon referred to as xenophagy [7]. Bacteria can mimic autophagy ingredient regulators to shut down their functions or directly bind to and recruit autophagy proteins to promote growth [8]. Autophagy can selectively target intracellular bacteria in the cytosol or vacuoles to impede their proliferation [9]. Typically, autophagosomes adorned with LC3 are generated to encapsulate the targeted bacteria and transport them to the lysosome for destruction [10]. Recent research has placed a lot of effort into researching how bacterial factors and ATG proteins interact, and this will continue for a long period [11]. By understanding how bacteria govern autophagy, scientists can develop better medications and cure bacterial diseases more efficiently [12]. In this review, we deliver an introduction to how autophagy works, and we focus on how bacteria and autophagy interact, including how bacteria are targeted for removal by autophagy and how bacteria can stop this process or use it to stay alive.

## 2. Molecular Mechanism of Autophagy in Perspective with Bacterial Pathogenesis

Autophagy is a fundamental and highly resilient cellular mechanism that recycles damaged proteins or organelles to preserve cell homeostasis [13]. It protects the host from intracellular pathogens such as *Mycobacterium tuberculosis*, *Salmonella typhimurium*, *Listeria monocytogenes* and *Staphylococcus aureus* and plays a crucial function in cellular homeostasis [14]. In a mechanism known as xenophagy, they can be directly targeted in the cytosol, or by Light Chain 3-associated phagocytosis (LAP), they can be targeted in vacuoles and phagosomes. While the bacteria are being trapped within the phagosome, LAP activates the autophagic apparatus. Autophagy can, therefore, function as an innate immune defense mechanism to combat bacterial infection [15,16,17]. While certain microorganisms have evolved strategies to take advantage of autophagy within the cell’s interior, other have developed intriguing techniques to avoid detection, disrupt the autophagic process, or escape autophagy (Figure 1) [13].

### 2.1. Mycobacterium tuberculosis

The etiological cause of tuberculosis (TB), perhaps one of the ancient human infections and with one of the top 10 mortality rates globally, is *Mycobacterium tuberculosis* [18]. The capacity of *Mycobacterium tuberculosis* to multiply and survive inside alveolar macrophages by interfering with phagolysosome formation is one of the critical aspects of TB pathogenesis [19,20]. In the past twelve years, autophagy has emerged as a crucial defense mechanism that the host uses to prevent the dissemination of *Mycobacterium tuberculosis* [19]. By controlling the transcription of nearly two thousand genes, interferon-γ (IFN-γ) is crucial for protection against infection [21]. IFN-γ, a potent macrophage activator, was able to imitate the impacts of rapamycin on the activation of autophagy employing immunity-related p47 guanosine triphosphatases (IRG) Irgm1 (LRG-47) [19]. These findings place autophagy at the core of the immune systems that help the body fight off an infection with *M. tuberculosis*. After making their initial discoveries, the same group proceeded to show that Irgm1 and IRGM (Immunity-related GTPase family M) protein, a human ortholog, are both required for the initiation of autophagy, resulting in large autolysosomes that assist with the intracellular limitation of the growth of *M. tuberculosis* after macrophage activation by IFN-γ [19]. Interferon-induced guanylate-binding protein (GBP), which is elevated in the presence of the cytokine, plays a function in the involvement of IFN-γ in autophagy. GBPs have been shown to facilitate oxidative killing and the transfer of antimicrobial peptides to autophagolysosomes, which aid in regulating *Mycobacterium tuberculosis* intracellular multiplication [22]. Together, these research findings showed that xenophagy and its activation by IFN-γ play a crucial role in vitro in regulating *Mycobacterium tuberculosis* intracellular proliferation.

According to several recent research studies, *M. tuberculosis* employs complex methods to evade xenophagy and multiply within host cells. *M. tuberculosis* promotes the development of various types of microRNAs (miRNAs) to evade xenophagy by interfering with several elements of cellular physiology, in addition to miR33 (microRNAs 33) and miR33 expression to alter cellular energy utilization and metabolism stages [23] and miRNA125a to block UVRAG (UV Radiation Resistance Associated Gene) expression [24] (Figure 2). In this instance, *M. tuberculosis* infection causes an increase in the expression of miR30A, which lowers Beclin-1 levels of expression and inhibits autophagosome elongation, promoting *M. tuberculosis* intracellular survival [25].

### 2.2. Salmonella typhimurium

Gram-negative, non-spore-forming intracellular *S. typhimurium* bacteria can trigger gastroenteritis [26]. The SipB (Salmonella Invasion Protein B) protein from *S. Typhimurium* induces autophagy, which kills macrophages [27]. Salmonella-containing vacuoles (SCVs) are ruptured by the *S. Typhimurium* type III secretion system (TTSS) and the bacteria inside these destroyed SCVs are then the target of autophagy. Ubiquitination occurs in these autophagy-targeted microorganisms [28]. Due to damage to the SCV membrane caused by *S. typhimurium* infection, amino acid deprivation occurs in epithelial cells, which then prompts the activation of autophagy [29]. When it was observed to coexist with *S. typhimurium,* SQSTM1 (Sequestosome-1), an autophagy adaptor, was discovered to be connected to bacterial autophagy [30]. According to research, NDP52 (Nuclear Dot Protein 52 kDa), another adaptor, is involved in *S. typhimurium’s* autophagy [31]. NDP52 recruits TBK1 (TANK-binding kinase 1) to ubiquitinate *S. typhimurium* after binding with the adaptor proteins Sintbad and Nap1 (Nucleoid Associated Protein 1). This causes an innate immune reaction that is autonomous. NDP52 and SQSTM1 are independently enlisted to *S. typhimurium* with the same kinetics, and the loss of either adaptor impairs autophagy. Together, NDP52 and SQSTM1 promote effective antimicrobial autophagy [32]. Additionally, *S. typhimurium* has developed ways to avoid autophagy. In the cultured cell, *S. typhimurium* leads to the development of ubiquitinated aggregates, but *S. typhimurium*’s SseL deubiquitinates SQSTM1-bound proteins, which are virulence proteins observed in these aggregates that minimize autophagic activity and promote bacterial replication [33] (Figure 3). By inhibiting autophagy through the Akt/PI3K-mTOR (Phosphotidylinositol-3-kinase-mammalin target of rapamycin) regulatory pathway, the non-receptor tyrosine kinase focal adhesion kinase supports the intracellular proliferation of *S. typhimurium* in macrophages [34].

### 2.3. Listeria monocytogenes

Listeriosis is caused by the Gram-positive, facultatively anaerobic rod-shaped bacteria *L. monocytogenes* [35]. When *L. monocytogenes* expresses listeriolysin O (LLO), which is a pore-forming toxin, autophagy is induced, although phospholipases A and B are unnecessary for this to happen [36]. The diaminopimelic acid-type peptidoglycans on *L. monocytogenes* are recognized by the Drosophila Peptidoglycan Recognition Protein LE (PGRP-LE), which causes autophagy to be induced and limits *L. monocytogenes’* intracellular proliferation [37]. It has also been documented that Pattern Recognition Receptors (PRRs) contribute to the activation of autophagy following *L. monocytogenes* infection. When *L. monocytogenes* is present, Nod (Nucleotide oligomerization domain)-like receptors 1 and 2 and Toll-like receptor 2 work through the downstream extracellular signal-regulated kinases to activate the autophagic response [38]. Although listeriosis is often a foodborne illness, it can be harmful for those who have compromised immune systems and for pregnant women. The 2/3 (Arp2/3) complex, an action-related protein, is activated by the production of ActA, a surface protein that causes actin to reorganize and the cytosolic bacterium to become motile [39]. Act A is crucial for escaping autophagy because it inhibits ubiquitination and the recruitment of the proteins that trigger autophagy [40] (Figure 4).

### 2.4. Staphylococcus aureus

The Gram-positive, non-motile cocci-shaped bacterium *Staphylococcus aureus* is responsible for several life-threatening conditions, such as sepsis, endocarditis, and pneumonia. Autophagy does not increase *S. aureus* burden in bloodstream and lung infections in vivo, despite early tissue culture experiments using HeLa cells highlighting how *S. aureus* hijacks the process to increase bacterial survival [41]. Instead, a potent pore-forming toxin (α-toxin) released by *Staphylococcus aureus* causes cell death by binding to ADAM10 (A Disintegrin and Metalloproteinase Domain containing protein 10) on the outermost surface of desired cells like the endothelium. The autophagy protein ATG16L1 and various autophagy elements protect host cells from these lethal effects [42]. The ATG16L1 complex increases cell survival by downregulating ADAM10 levels, which reduces the receptor number necessary for α-toxin binding. Due to its role in the host resilience response or tolerance, the autophagy mechanism avoids excessive tissue damage in this setting [43]. By activating signaling pathways or proteins, *S. aureus* can avoid autophagy. The PI3K/protein kinase B (AKT)-Beclin1 signaling pathway is strongly connected to the autophagy activated by *S. aureus* infection of the murine macrophage cell line RAW264.7 [44] (Figure 5). The degree of *S. aureus*-induced autophagy is dramatically reduced after inhibiting this signaling pathway [44]. These findings demonstrate that *S. aureus*-infected cells produce autophagosomes that cannot combine with lysosomes, leading to autophagy. As a result, *S. aureus* may grow in the autophagosome without destruction, inhibiting the autophagosomes from becoming acidified [41].

## 3. Interplay between Autophagy and Bacterial Toxins

### 3.1. Interactions between Host Cell Autophagy and Bacterial Toxins Involved in Pathogenesis

Autophagy is a fascinating process that plays a crucial role in the body’s defense against intracellular bacteria. Bacterial toxins intricately interact with the autophagic pathway, which is tightly regulated [45]. Many virulence factors and toxins produced by intracellular bacterial pathogens can restrict autophagy [46]. These bacterial products include lipopolysaccharides (LPS), pore-forming toxins from *Streptococcus pyogenes* and *Listeria monocytogenes*, membrane-spanning secretion systems from *Mycobacterium tuberculosis* and *Salmonella* serovar *Typhimurium*, and bacterial adenylate cyclases. Host cells can initiate autophagy during bacterial infection to eliminate intracellular pathogens and/or toxins [47].

#### 3.1.1. Lipopolysaccharide (Endotoxin)

Lipopolysaccharide (LPS), a common molecule on the surface of Gram-negative bacteria, is recognized by human innate immune cells [48]. The term endotoxin is interchangeably used with LPS [49]. One of the most immunostimulatory elements of Gram-negative bacteria’s outer membrane is LPS, which can interfere with the autophagy system [47]. During sepsis, endotoxin-induced autophagy maintains lipid metabolism homeostasis [50]. The endotoxin produced by the many bacteria in the gut can kill the host multiple times. Large amounts of highly toxic and immune-active debris, including peptidoglycan and lipopolysaccharide (LPS), are produced after the death of these bacteria [51]. By eliminating microorganisms and reducing endotoxin-induced inflammatory reactions in the intestinal epithelium, selective autophagy helps to maintain intestinal homeostasis [52].

It has been proposed that LPS promotes autophagy, a defensive mechanism activated in peritoneal mesothelial cells in response to *E. coli* infection [53]. Intestinal homeostasis can be maintained by autophagy, which can reduce endotoxin-induced inflammatory responses in the intestinal epithelium [54]. An essential part of the autophagic system that regulates endotoxins’ inflammatory, immunological response is autophagy-related 16-like 1 (Atg16L1) [55]. Atg16L1 regulates endotoxin-induced inflammatory pathway activation. Atg16L1 is required to form the Atg12-Atg5 conjugate and defines the site where microtubule-associated protein 1 light chain 3 (LC3) is attached to phosphatidylethanolamine (PE), which is an essential process for autophagy by engaging the Atg3-LC3 intermediate to a source membrane of an autophagosome. Consequently, both autophagosome formation and degradation of long-lived proteins happen in Atg16L1-containing cells [55] (Figure 6).

#### 3.1.2. Bacterial Pore-Forming Toxins (PFTs)

Pore-forming toxins (PFTs) comprise most bacterial protein toxins, which are crucial virulence components produced by many pathogenic bacteria (Table 1). The bacterium secretes these toxins, which multimerize once they attach to a membrane receptor, creating an amphipathic structure that acts as a pore. *Streptococcus*, *Staphylococcus*, *Bacillus*, *Listeria*, and *Clostridium* are the few bacteria that make the comprehensive family of pore-forming toxins. A standard method of attack is to create pores in the target cells’ membranes, which results in an imbalance of cellular ions [56].

#### 3.1.3. Non-Pore-Forming Toxin (AB Toxin)

Several related proteins that do not entirely meet the criteria to be PFTs are called “AB” toxins, for example, anthrax toxin from *Bacillus anthracis*. The translocation domains of certain AB toxins are most notable where the B subunit is accountable for attaching to the target cell and translocating the A subunit into the cytoplasm, which has enzymatic activity [47].

#### 3.1.4. Bacterial Adenylate Cyclases (ACs)

Cyclic AMP regulates many cellular processes, like pro-inflammatory cytokine production [61]. Many bacteria like *Pseudomonas aeruginosa*, *Yersinia pestis*, *Mycobacterium tuberculosis*, *Bacillus anthracis*, and *Bordetella pertussis* that produce adenylate cyclase toxins have been found to suppress autophagy stimulation by increasing the levels of cAMP and are supposed to down-regulate the autophagy in host cells [47]. For instance, adenylyl cyclase, an edema factor toxin from *Bacillus anthracis*, can increase intracellular cAMP levels directly and cholera toxin from *Vibrio cholerae* functions as an ADP-ribosyltransferase that can increase cAMP indirectly by triggering ACs. These two toxins inhibit the stimulation of host cell autophagy [61]. The ADP-ribosyltransferases produced by another pathogen, such as *Bordetella pertussis*, can indirectly activate host ACs. *Staphylococcus aureus* may also synthesize adenosine, increasing cAMP levels, which it uses to stimulate G-protein-coupled adenosine receptors in host cells [62].

### 3.2. Autophagy Evasion by a Bacterial Toxin

#### Evasion by Pore-Forming Toxin

Pathogens have developed a variety of tactics to obstruct autophagic signaling pathways and prevent autophagosome–lysosome fusion to form autolysosomes. Group A *Streptococcus* (GAS) has developed several ways to prevent autophagy from persisting in the cell for a long time. For example, they use certain virulence factors, such as *Streptococcus* pyrogenic exotoxin B (SpeB) and *Streptolysin* O (SLO), to avoid autophagy degradation [60,63], as shown in Figure 7.

## 4. Pharmacological Induction of Autophagy Targeting Bacterial Pathogen

Pharmacologically inducing autophagy has been proposed as a novel approach for targeting bacterial pathogens. Autophagy has been found to aid in removing intracellular bacteria like *Salmonella* and *Mycobacterium tuberculosis*, and certain drugs like rapamycin and its derivatives have been demonstrated to boost autophagy and aid in bacterial pathogen elimination [64]. For instance, rapamycin treatment enhanced the clearance of *Salmonella typhimurium* in macrophages by inducing autophagy [65]. Similarly, rapamycin treatment increased the autophagy-mediated release of *Mycobacterium tuberculosis* in macrophages [66]. These findings suggest that pharmacological induction of autophagy could be a promising approach for targeting bacterial pathogens.

### 4.1. Synthetic Drug

#### 4.1.1. Autophagy Is Triggered by mTOR Signaling Inhibitors

mTOR regulates intracellular processes such as protein translation, autophagy, and metabolism [67]. It consists of two different complexes, mTORC1 and mTORC2, which perform distinct functions [68]. One crucial Ser/Thr kinase that regulates autophagy is Unc-51, which can act as both an effector and a negative regulator of mTORC1. Inhibitors of mTORC1, such as rapamycin, can affect the activity of Unc-51 and thereby influence autophagy [69]. Understanding the complex interplay between mTOR and its downstream effectors is crucial for developing new treatments for diseases related to abnormal autophagy or metabolism [70]. Several studies have demonstrated the ability of rapamycin to induce the autophagy-mediated clearance of various pathogens, including bacteria, viruses, and parasites [71]. For example, a study found that rapamycin treatment enhanced the autophagic release of *Mycobacterium tuberculosis* in macrophages [72]. Furthermore, another study found that rapamycin treatment enhanced the clearance of *Toxoplasma gondii*, a protozoan parasite, in infected cells through the induction of autophagy [73].

#### 4.1.2. Activators of AMPK Activate Autophagy

AMP-activated protein kinase (AMPK) is a critical energy-sensing protein that regulates various metabolic processes, including glucose uptake and fatty acid oxidation. AMPK activation has been linked to autophagy, a cellular process that degrades damaged organelles and pathogens [74]. Studies have shown that AMPK activation by metformin, a commonly used anti-diabetic drug, can stimulate the autophagy of pathogens. For example, metformin treatment has been shown to enhance the autophagy of *Mycobacterium tuberculosis* in human macrophages [75]. In addition, metformin has been shown to induce the autophagy of the protozoan parasite *Toxoplasma gondii*, leading to its clearance in infected cells [76]. These findings suggest that AMPK activators such as metformin can promote the autophagy-mediated clearance of pathogens, potentially providing a new avenue for developing antimicrobial therapies.

#### 4.1.3. Autophagy Is Triggered by Blockers of Class I PI3K Signaling

Class I phosphoinositide 3-kinases (PI3Ks) are involved in many cellular processes, including cell proliferation, survival, and metabolism. AKT is a vital mediator of class I PI3K signaling, and inhibitors like perifosine can activate autophagy [77]. Furthermore, autophagy can be activated by inhibiting receptor tyrosine kinases that inhibit AKT, thereby inhibiting mTORC1 or regulating beclin-1. The interplay between AKT, PI3K, mTORC1, and beclin-1 is complex, and understanding these relationships is critical for developing effective treatments for diseases associated with autophagy dysfunction [78]. In addition, class I PI3K signaling has been implicated in regulating autophagy, a cellular process that degrades intracellular components, including invading pathogens. Inhibitors of class I PI3K signaling can activate the autophagy of pathogens by suppressing the activity of the mTOR pathway, a key negative regulator of autophagy. When mTOR activity is inhibited, autophagy is activated, leading to the sequestration and degradation of intracellular components, including pathogens. Studies have shown that the inhibition of class I PI3K signaling can enhance the autophagy-mediated clearance of various pathogens, including bacteria and viruses. For example, a study found that treatment with a class I PI3K inhibitor enhanced intracellular bacteria clearance in macrophages through autophagy activation [79]. Furthermore, a study found that inhibition of class I PI3K signaling enhanced the autophagy-mediated clearance of hepatitis B virus (HBV) in liver cells. The article suggested that class I PI3K signaling inhibition could be a potential therapeutic strategy for HBV infection [80].

#### 4.1.4. Autophagy Is Triggered by Inhibitors of Inositol Mono-Phosphatase

Lithium, a well-known mood stabilizer, has been reported to induce the autophagy of pathogens, including bacteria and viruses, in various cell types. Lithium-induced autophagy is mediated by inhibiting inositol mono-phosphatase (IMPase) [81]. This enzyme catalyzes the conversion of inositol monophosphate to inositol, leading to the depletion of inositol and the activation of autophagy-related signaling pathways. The reduction of inositol caused by lithium has been shown to inhibit the production of the secondary messenger inositol-1,4,5-trisphosphate (IP3) and the subsequent release of Ca^2+^ from the endoplasmic reticulum, resulting in the activation of the AMPK-mTOR signaling pathway, which in turn induces autophagy. Furthermore, lithium has been shown to induce the autophagy of *Mycobacterium tuberculosis* by upregulating the expression of autophagy-related genes, including Beclin-1 and LC3, and downregulating the expression of virulence-associated genes. These findings suggest lithium could be a potential therapeutic agent for treating infectious diseases [82,83,84]. Table 2 lists the synthetic medications that cause microorganisms to undergo autophagy.

### 4.2. Natural Compounds

The discovery of natural chemicals that target autophagy is an intriguing new dimension in the field of medicine that has the potential to have a long-lasting influence on the fight against bacterial infections. The autophagy of pathogens can be induced by natural substances, which are listed in Table 3.

#### 4.2.1. Curcumin

Turmeric polyphenol, also known as curcumin, has been shown to induce autophagy in various pathogens. One study found that curcumin treatment increased the expression of autophagy-related genes and the number of autophagic vacuoles in *Staphylococcus aureus*-infected cells, reducing bacterial burden and increasing host survival [93]. Another study demonstrated that curcumin-induced autophagy in *Leishmania donovani*-infected macrophages was associated with inhibiting the parasite’s intracellular survival and enhancing the host immune response [94]. The exact mechanism by which curcumin induces autophagy in pathogens is not yet fully understood. However, it is believed that curcumin may activate autophagy by regulating signaling pathways, such as the AMPK/mTOR pathway [95]. Additionally, curcumin has been shown to disrupt the integrity of bacterial membranes, releasing cytoplasmic contents and triggering an autophagic response [93].

#### 4.2.2. Chloroquine

Chloroquine, an anti-malaria drug, induces the autophagy of pathogens by inhibiting lysosomal function and impairing the fusion of autophagosomes with lysosomes. This results in the accumulation of autophagic vacuoles and the eventual death of the pathogen. Chloroquine has been shown to induce autophagy in various pathogens, including *Plasmodium falciparum*, *Toxoplasma gondii*, and *Mycobacterium tuberculosis* [96,97]. However, recent studies have suggested that chloroquine may have other mechanisms of action beyond its effects on autophagy, and its clinical efficacy is still being investigated.

#### 4.2.3. Chrysin

Chrysin is a flavonoid with various biological properties, including the induction of autophagy in different cell types. Studies have shown that chrysin can induce autophagy in pathogenic bacteria by upregulating autophagy-related genes, resulting in the clearance of bacterial cells [98]. One study demonstrated that chrysin-treated *Pseudomonas aeruginosa*-infected macrophages showed increased autophagy activity, significantly reducing bacterial load. The study suggests that chrysin could be a promising candidate for developing novel antimicrobial therapies based on autophagy modulation [99].

#### 4.2.4. Oridonin

One study investigated the effect of oridonin on *Mycobacterium tuberculosis* (Mtb) infection in macrophages and found that oridonin can induce autophagy and enhance the clearance of Mtb in infected cells. Moreover, scientists have discovered that oridonin is capable of inducing autophagy in Mtb-infected macrophages by activating AMP-activated protein kinase (AMPK) and suppressing mammalian target of rapamycin complex 1 (mTORC1) [100]. It was discovered in a separate study that oridonin can stimulate autophagy in *Toxoplasma gondii*-infected HL-60 human promyelocytic leukemia cells; *Toxoplasma gondii* is a type of protozoan parasite known to cause toxoplasmosis. In addition, the researchers found that oridonin can increase the expression of Beclin-1 and LC3-II, two key autophagy markers, and enhance the clearance of *T. gondii* in infected cells [101].

#### 4.2.5. Quercetin

Quercetin, present in different fruits, vegetables, and herbs, possesses anti-inflammatory and antioxidant characteristics due to its flavonoid nature. It has been shown to induce autophagy in various pathogens, including bacteria and viruses. In one study, Quercetin induced the autophagy of the pathogenic bacterium *Salmonella typhimurium* in human cells by upregulating the expression of autophagy-related genes and increasing the number of autophagosomes in infected cells [102]. Similarly, in another study, Quercetin was found to induce autophagy of the hepatitis C virus by inhibiting the PI3K/AKT/mTOR pathway [103]. These findings suggest that Quercetin has the potential to induce the autophagy of various pathogens and may have implications for the treatment of infectious diseases.
biomedicines-12-01757-t003_Table 3Table 3Natural compounds that induce the autophagy of pathogens.MechanismCompoundsReferenceActivation of AMPK; inhibition of mTORC1Curcumin[95]Inhibiting lysosomal functionChloroquine[96,97]Upregulating autophagy-related genesChrysin[98]Inhibition of the AKT/mTOR signaling pathwayOridonin[100]Inhibiting the PI3K/Akt/mTOR signaling pathwayQuercetin[102]Inhibiting the PI3K/Akt/mTOR pathwayMorusin[104]Activating the TLR4-mediated signaling pathwayPaclitaxel[105]Upregulating the AMPK/mTOR signaling pathwayOleanolic Acid[106]

## 5. Recent Advances and Updates on Autophagy Signaling in Bacterial Infection

Recent findings have shown that the autophagy procedure is complex and might be different depending on the pathogenic strain. These findings also indicate that autophagy plays a crucial role in both human immunity and illness, which makes it desirable target for advanced therapeutic strategies [107]. In tuberculosis patients, autophagy is a host defense mechanism against the intracellular pathogenic bacteria *Mycobacterium tuberculosis*. Patients with *M. tuberculosis* strains with lower autophagy-activating capacities have more severe illnesses and show worse clinical outcomes [108]. This study also showed that among “ancient” and “modern” strains, modern strains can bypass the autophagy mechanism by modulating T-cell responses [109]. Some experiments have found that highly pathogenic *M. tuberculosis* may frequently use host non-coding microRNAs (miRNAs) to improve pathogenicity by limiting host-mediated antimicrobial signaling pathways. Likewise, host-persuaded miRNAs and long non-coding RNA (lncRNAs) boost autophagy to reduce bacterial growth [110].

Another intercellular bacterium, *Salmonella enterica* serovar Typhimurium, has evolved ways to avoid or manipulate autophagy by direct interactions between autophagy proteins and the effector proteins released by the infectious agent during pathogenesis. This study demonstrated through a computational network analysis approach that the SPI-1 effector protein *SpoE*, from *Salmonella* Pathogenicity Island-1, directly binds to SP1, the host transcription factor. After this interaction, the autophagy-related gene MAP1LC3B’s expression is controlled adversely by *SpoE.* This research demonstrated that *SopE* may play two distinct roles in the regulation of autophagy: after the initial rise in MAP1LC3B transcription brought on by *Salmonella* infection, the subsequent decrease in MAP1LC3B transcription at 6 h after infection was reliant on Sop. *SopE* is also thought to impact the autophagy inflow since it is retained at the membrane of the *Salmonella*-containing vacuole “(SCV)” for a few hours after Salmonella has been internalized. MAP1LC3 attaches the foreign substance to the wall of the vehicle using the SQSTM1/p62 adaptor protein after being lapidated and brought to the membrane formation site. The study opted for the autophagosome-linked molecules MAP1LC3 and SQSTM1/p62 because they are reliable indicators of the autophagy inflow in mammalian cells [111].

The recently developed “LC3-associated phagocytosis (LAP)” is an immensely bactericidal non-canonical autophagy pathway that enhances the degradation of bacteria inside LAPosomes against bacterial infections. Another currently identified non-canonical autophagy pathway, known as pore-forming toxin-induced non-canonical autophagy (PINCA), initiates autophagy when phagosomal damage is introduced by the pore-forming toxins produced by bacteria. PINCA, like other non-canonical pathways, does not need the production of NADPH oxidase 2 (Nox2)-derived ROS, which is necessary for LAP’s LC3 phagosome decoration, and it also does not need the ULK complex. As a result, PINCA activation in Nox2-deficient peritoneal macrophages and wild-type bone-marrow-derived macrophages cannot generate enough ROS to trigger LAP because of insufficient Nox2 expression [112,113]. To cause LC3 recruitment to the injured phagosomes, or PINCA, the damage caused by the pore-forming toxin listeriolysin O (LLO) of *Listeria monocytogenes* or by the various pore-forming toxins of *S. aureus* was required. When pathogens evade LAP or PINCA, they can be recaptured by xenophagy. Further, xenophagy can also be evaded by pathogenic bacteria [114].

*Klebsiella pneumoniae* infection of the lungs led to inadequate, unfit autophagy in the lungs. In this research, *K. pneumoniae* is administered into the lungs of mice through cannulation, where systemic infection demonstrates sepsis development [115]. By forcing the production of the active, altered Becn1F121A or by administering a Beclin-1-operating peptide, Tat-beclin-1 (TB-peptide), Beclin1 is activated in mice, which increased autophagy and greatly enhanced sepsis outcomes, including declines in illness ratings, infection spread, and inflammatory processes. It is clear that increased Beclin-1 signaling reduced lung injury by lowering the levels of alveolar blockage, bleeding, inflammatory cell penetration, and alveolar wall depth [116]. This study demonstrates the clinical efficacy of activating Beclin-1 to treat pneumonia-induced sepsis by modulating autophagy.

*Staphylococcus aureus* can cause autophagy while infecting the host as an opportunistic intracellular infectious agent and following degradation of the ubiquitin-coated intracellular pathogens mediated by “Sequestosome 1 (SQSTM1/p62)”, “nuclear domain protein 52 (NDP52/CALCOCO2)”, and “optineurin (OPTN)” autophagy receptors. These receptors meet the autophagosomal membrane-coupled protein LC3 to bind ubiquitin to microorganisms for capturing pathogens in self-destructing vesicles [117,118]. Nevertheless, *S. aureus* has created ways to bypass the autophagy pathway [13]. In murine fibroblasts, activation of “mitogen-activated protein kinase 14 (MAPK14)” and “ATG5” by *S. aureus* has been shown to prevent autophagosome maturation [117]. A recent study showed that *S. aureus* induced the development of autophagosomes in bovine mammary epithelial cells to aid in intracellular reproduction. Some studies also confirmed that *S. aureus* intracellular multiplication in Chinese hamster ovary cells might be inhibited by protein kinase C (PKC) overexpression suppressing autophagy [119]. Figure 8 illustrates the latest advancements in autophagy within various bacterial cellular and molecular mechanisms.

## 6. Limitations and Overcoming Bacterial Defense Mechanisms via Inducing Autophagy

Most intracellular bacterial pathogens have evolved to avoid or exploit xenophagy, a significant hurdle. Cell culture investigations and inbred mouse models have shown that xenophagy does not control bacterial infections. Whether stimulating xenophagy will overcome these protections clinically or whether addressing bacterial defensive mechanisms is needed is unknown. Since cell culture phenotypes have been modest and do not necessarily match well with the in vivo results, it is also crucial to determine if cell culture trials are an accurate model for testing autophagy targeting efficacy. Alternative therapeutic approaches could investigate autophagy protein activities outside of xenophagy that may affect infection outcomes and the synergy between autophagy modulators and antibiotics. Like targeting xenophagy, these regions need further investigation to determine their clinical relevance but may hold promise for future therapeutic techniques.

To create a prolonged infection, several bacterial pathogens have gained the ability to infiltrate host cells or suppress host autophagy. There have been recent breakthroughs in studying the interplay between antibacterial autophagy (xenophagy) and several bacterial infections. Host cells express pattern recognition receptors to recognize the pathogen-associated molecular patterns of diverse microorganisms during infection. Innate receptors can activate an intracellular signaling cascade to activate antimicrobial effector mechanisms to clear infections by identifying alien or dangerous chemicals. Autophagy is one of the effector pathways downstream of these receptors and is crucial to innate and adaptive pathogen immunity. Autophagy degrades microorganisms and activates protective mechanisms such lysozyme production, the ubiquitin-mediated pathway, and antigen presentation through these interactions. Recent findings show autophagy regulates innate immunity as a “tuning module” to prevent excessive inflammatory responses and inflammasome activation. It is necessary for the bulk breakdown of cytoplasmic components within lysosomes that autophagy works as a defensive mechanism against multiple invading microorganisms and that there are specialized roles and regulatory mechanisms of autophagy in association with innate immune pathways during infection.

## 7. Conclusions

Autophagy affects several immunological functions, including the inflammatory process, phagocytosis, antigen presentation, and the release of bactericidal agents, in addition to the direct killing of bacteria in autophagosomes [120]. Recent in vitro, in vivo, and molecular investigations show selective autophagic breakdown of bacteria and viruses [7]. The cell’s defense reaction to intracellular microorganisms has also been rapidly understood. Many bacterial infections have several mechanisms to bypass host defense responses and survive and replicate [121]. Autophagy pathways work with innate immunity to fight infections. Several immune effectors regulate autophagic reactions to bacterial infection. The host autophagy system manipulates autophagic adaptors or cargo receptors to target bacteria, proteins, and damaged organelles [8]. Novel antibacterial drugs will be developed when host autophagy and microbial pathogens are revealed. It is difficult to evaluate whether autophagy activation or inhibition promotes infection due to the complexity of autophagy-mediated immune regulation and the flexibility of bacterial autophagy methods. Thus, in vivo investigations will help us understand macrophage autophagy and bacterial infection and develop better treatments for troublesome conditions.

## Figures and Tables

**Figure 1 biomedicines-12-01757-f001:**
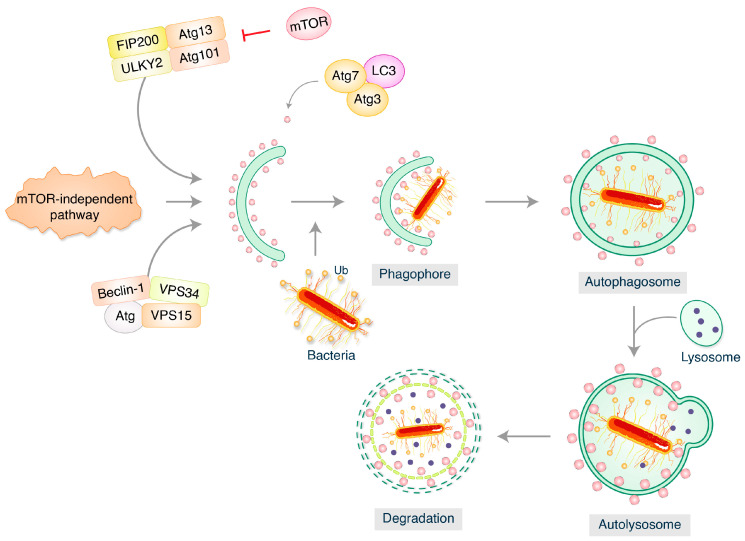
Bacterial pathogens and autophagy. Autophagy requires activation of the complex between uncoordinated 51-like kinases 1 and 2 (ULK1–ULK2) and scaffold proteins ATG13, FIP200, and ATG101. Nucleation recruits’ proteins and lipids to the phagophore. Vesicles that dynamically enter and exit the phagophore contain the multi-spanning transmembrane protein ATG9.

**Figure 2 biomedicines-12-01757-f002:**
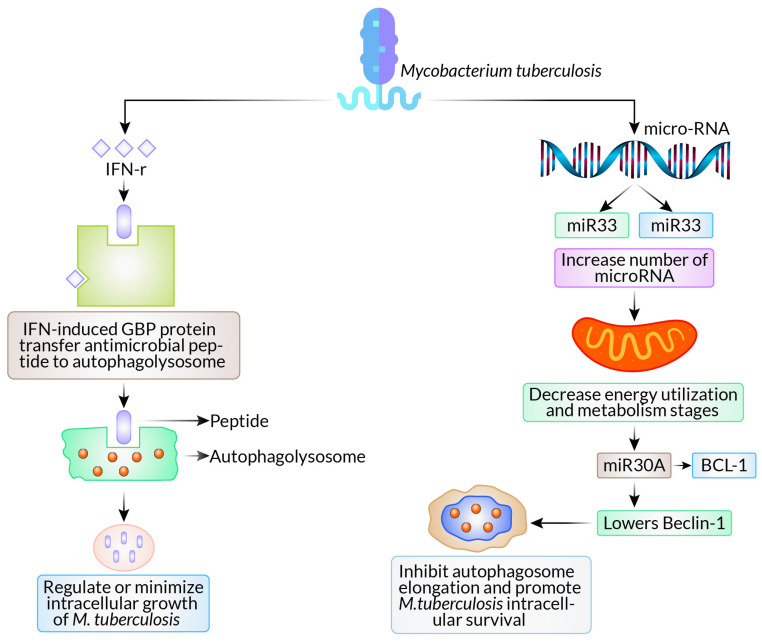
Autophagy as a defense mechanism and evasion of autophagy pathways in *Mycobacterium tuberculosis*.

**Figure 3 biomedicines-12-01757-f003:**
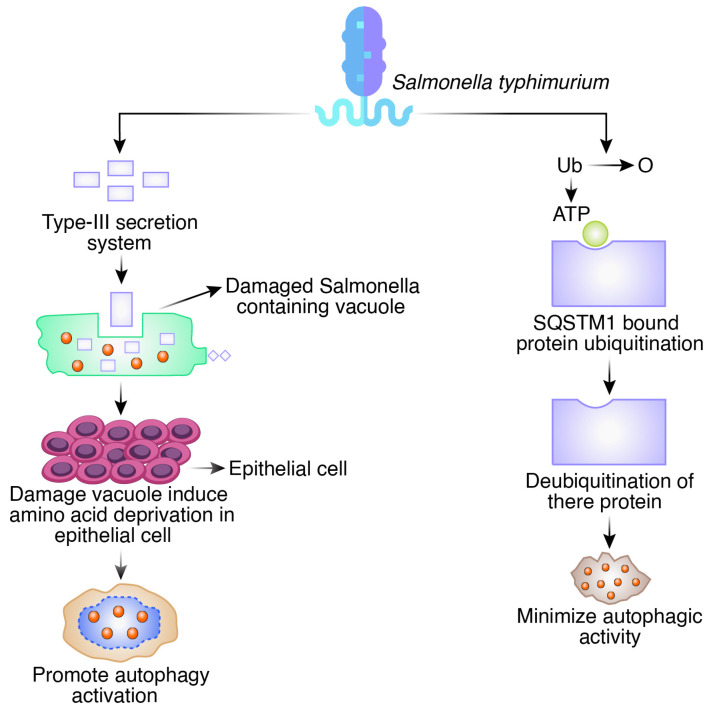
Autophagy as a defense mechanism and evasion of autophagy pathways in *Salmonella typhimurium*.

**Figure 4 biomedicines-12-01757-f004:**
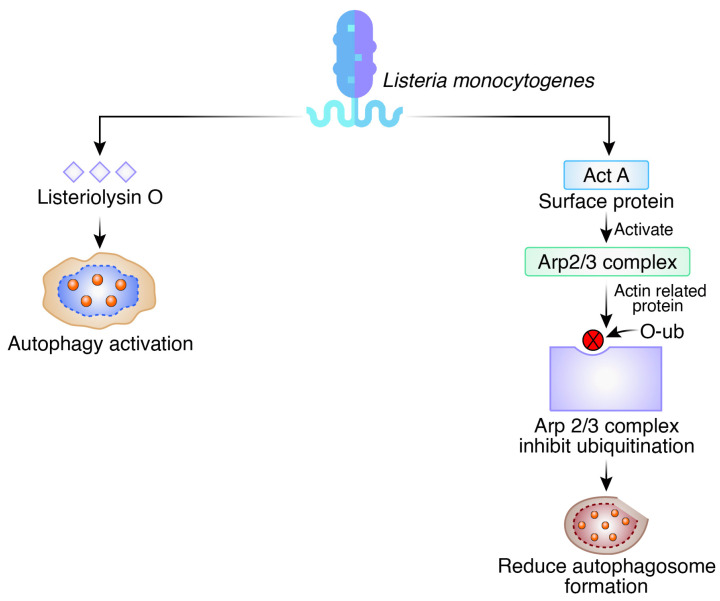
Autophagy as a defense mechanism and evasion of autophagy pathways in *Listeria monocytogenes*.

**Figure 5 biomedicines-12-01757-f005:**
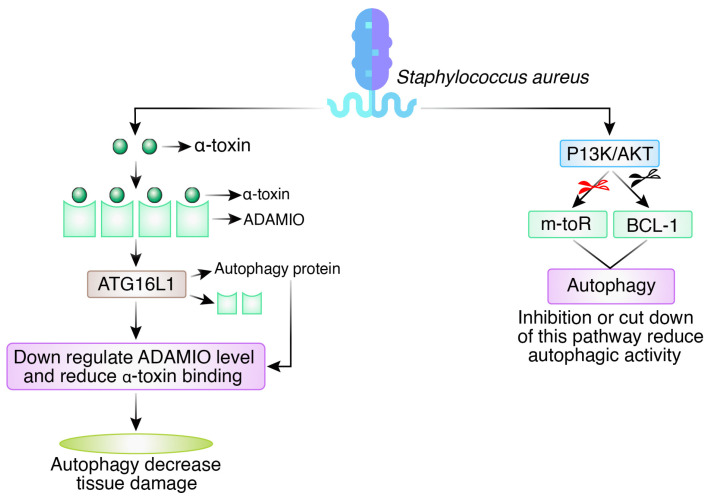
Autophagy as a defense mechanism and evasion of autophagy pathways in *Staphylococcus aureus*.

**Figure 6 biomedicines-12-01757-f006:**
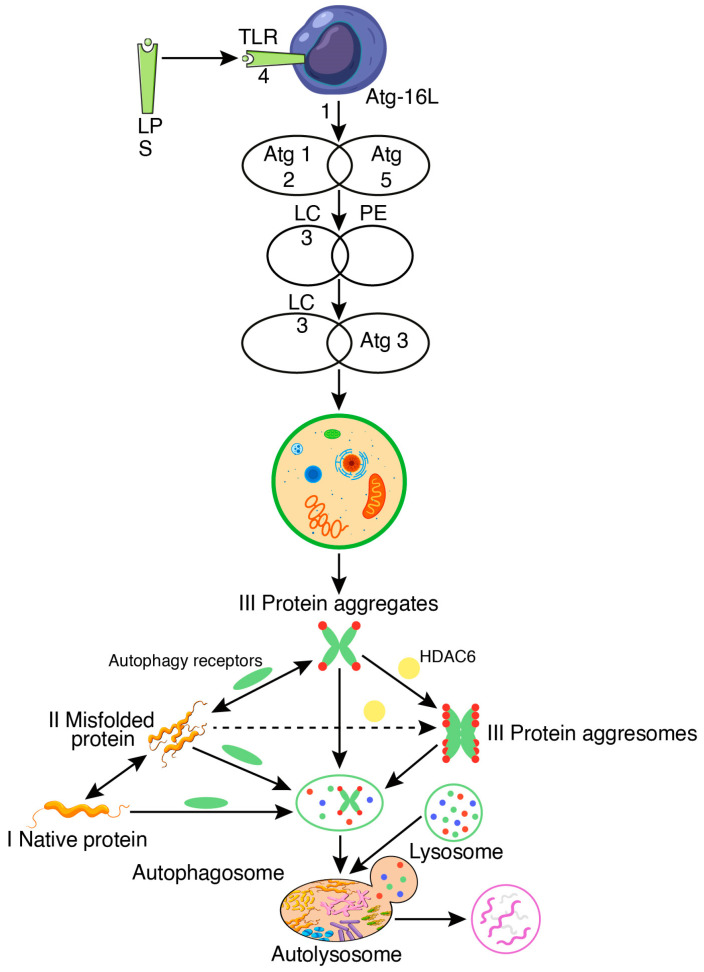
Regulation of the endotoxin-induced inflammatory response by autophagy.

**Figure 7 biomedicines-12-01757-f007:**
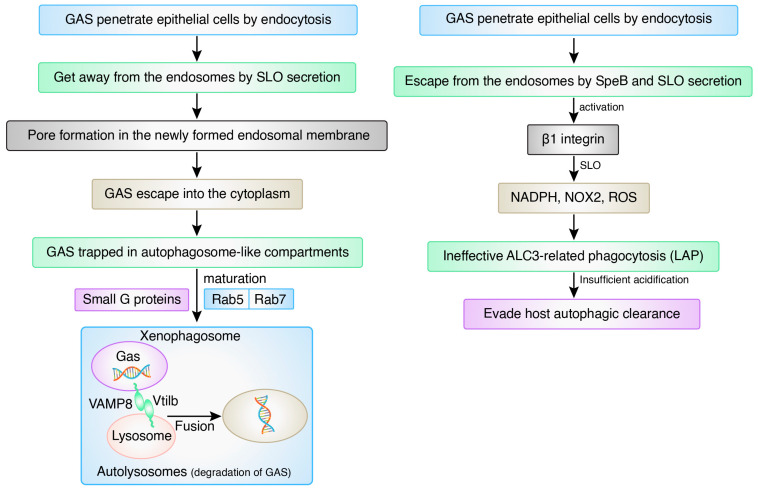
Outline of the action of autophagy after entering GAS.

**Figure 8 biomedicines-12-01757-f008:**
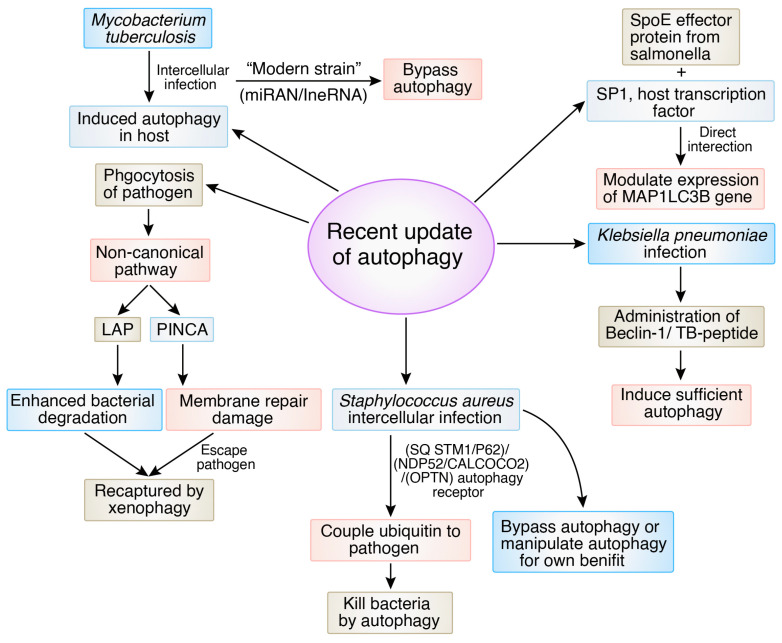
Recent updates of autophagy in different bacterial cellular and molecular mechanisms.

**Table 1 biomedicines-12-01757-t001:** Relationship between bacterial pore-forming toxins and autophagy.

Bacterium	Bacterial Toxin	The Function of Toxin in Host	The Link between Autophagy and Toxin	References
*Escherichia coli*	Colicins	Extensive vacuolation in epithelial cells.	In response to starvation, toxins activate the pathway that blocks translation and proceeds to autophagy.	[57]
*Vibrio cholerae*	Cytolysin (VCC)	Depending on the toxin dose and cell type, this toxin either forms vacuoles or causes lysis of the cell.	Vacuolization brought on by this exotoxin is connected to autophagy via the autophagic cell response after VCC intoxication.	[58]
*Streptococcus pyogenes*	Streptolysin O (SLO)	Induces AMP-activated protein kinase (AMPK) phosphorylation in epithelial cells.	Preventing TORC1 (target of rapamycin complex 1) AMPK inducing autophagy after decreasing the ATP/AMP ratio.	[57]
*Listeria monocytogenes*	Listeriolysin O (LLO)	Blocks phagosome–lysosome fusion by generating tiny pores and allowing *Listeria* to escape from phagosomes.	Targets damaged phagosomes to prevent bacterial escape and clear *Listeria monocytogenes* by activating autophagy genes.	[47,59]
*Staphylococcus aureus*	Alpha-toxin	Formation and binding of oligomers into lipid bilayers to form pores and decrease intracellular ATP levels.	Causes a drop in the intracellular cyclic adenosine monophosphate (cAMP) levels and favors a nontraditional autophagic process.	[47,60]
*Serratia marcescens*	Sh1A	Elicit an autophagic response in epithelial cells.	Unknown.	[47]
*Salmonella enterica* serovar *Typhimurium*	InvA and SipB	Damages *Salmonella*-containing vacuoles.	Restrict the intracellular growth by colocalizing with polyubiquitinated proteins.	[30,47]
*Helicobacter pylori*	VacA	Causes cytotoxic effects in impaired cells by vacuole formation.	Autophagy can degrade VacA in the early phase but this toxin impairs the autophagy process later.	[47]

**Table 2 biomedicines-12-01757-t002:** Synthetic drugs that induce autophagy of pathogens.

Mechanism	Compounds	References
Inhibition of mTORC1	Rapamycin	[85,86]
Activates AMPK	Metformin	[87,88]
Inhibitor of AKT pathway	Perifosine	[77]
Inhibiting inositol mono-phosphatase	Lithium	[82,83,84]
Inhibiting the function of L-type calcium channels	Verapamil	[89]
Inhibiting the mTORC1 pathway	Nimodipine	[90]
Upregulating the AMPK pathway	Nitrendipine	[91]
Inhibiting microtubule polymerization	Noscapine	[92]

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
