# Peer review of "An Update on the Study of the Molecular Mechanisms Involved in Autophagy during Bacterial Pathogenesis"

_biomedicines, 2024, doi:10.3390/biomedicines12081757_

Round 1
Reviewer 1 Report
Comments and Suggestions for Authors
A very interesting review, summarizing the current knowledge about autophagy during bacterial invasions. The authors described in detail the autophagy pathways of selected microorganisms, supplementing the text with interesting figures.
I have no objections to the substantive quality of the text, but I noticed a few editorial shortcomings:
- on line 248, the species name was incorrectly given: it should be Bacillus anthracis, because anthrax is the name of the disease
- there are excessive spaces or no spaces in the text (e.g. in lines: 50, 82, 121, 148, 167, 201,417, 511, etc.)
- several times in the Latin species name, both parts of the name are written in capital letters, while the first part of the name should be in capital letters and the second in small), e.g. lines: 95, 130, 147, 158, 281, etc.
- in Table I, all species are written with their full name, except for Helicobacter pylori, which is included with the shortened name "H. pylori" - this needs to be unified
- at the same time, in table I, at the end of text in some cells there are dots, in some there are commas, and in some there is no ending character. This needs to be standardized
- similarly, in tables 2 and 3, the text in the first column is either in capital or small letters - it needs to be unified
- in a few places, e.g. in figure 8 or line 427, 447, the Latin names are in a classic font, but they should be written in italic font
- in a few places, e.g. on line 424, the gene name is in a classic font, but should be written in italic font
And one last note - on line 465 the authors used the term "microbes". For me it is a popular term, not a scientific one. I think the more professional term will be "microorganisms".
I ask the authors to go through the text again and correct all these minor shortcomings.
Author Response
Reviewer 1
A very interesting review, summarizing the current knowledge about autophagy during bacterial invasions. The authors described in detail the autophagy pathways of selected microorganisms, supplementing the text with interesting figures.
>>Response: Thank you for your kind words and appreciation of our detailed assessment. We appreciate your thoughts and are glad you found the explanation of autophagy routes and bacterial invasions useful and supported by the figures. As you say, we try to explain complex biological processes clearly.
I have no objections to the substantive quality of the text, but I noticed a few editorial shortcomings:
- on line 248, the species name was incorrectly given: it should be Bacillus anthracis, because anthrax is the name of the disease
>>Response: We corrected accordingly in page 10 line 248.
- there are excessive spaces or no spaces in the text (e.g. in lines: 50, 82, 121, 148, 167, 201,417, 511, etc.)
>>Response: We checked and corrected spaces accordingly entire manuscript and marked with BLUE color.
- several times in the Latin species name, both parts of the name are written in capital letters, while the first part of the name should be in capital letters and the second in small), e.g. lines: 95, 130, 147, 158, 281, etc.
>>Response: We are grateful to the reviewer to raise this issue. We double checked and corrected accordingly and marked with BLUE color.
- in Table I, all species are written with their full name, except for Helicobacter pylori, which is included with the shortened name "H. pylori" - this needs to be unified
>>Response: We corrected accordingly in table page 9.
- at the same time, in table I, at the end of text in some cells there are dots, in some there are commas, and in some there is no ending character. This needs to be standardized
>>Response: We corrected accordingly in table page 9 all cells are dots.
- similarly, in tables 2 and 3, the text in the first column is either in capital or small letters - it needs to be unified
>>Response: We checked and modified tables 2 and 3 as per suggestion with capital letters.
- in a few places, e.g. in figure 8 or line 427, 447, the Latin names are in a classic font, but they should be written in italic font
>>Response: We checked and modified in figure 8 and others text.
- in a few places, e.g. on line 424, the gene name is in a classic font, but should be written in italic font
>>Response: We corrected.
And one last note - on line 465 the authors used the term "microbes". For me it is a popular term, not a scientific one. I think the more professional term will be "microorganisms".
I ask the authors to go through the text again and correct all these minor shortcomings.
>>Response: We checked and corrected the entire manuscript.
Reviewer 2 Report
Comments and Suggestions for Authors
I have read the Review entitled (An update on the study of the molecular mechanisms involved 2 in autophagy during bacterial pathogenesis) with a great attention. The review is well organized and updated with most recent references and discusses the whole situation thoroughly. I have no comments to the authors.
Comments on the Quality of English Language
I have read the Review entitled (An update on the study of the molecular mechanisms involved 2 in autophagy during bacterial pathogenesis) with a great attention. The review is well organized and updated with most recent references and discusses the whole situation thoroughly. I have no comments to the authors.
Author Response
Reviewer 2
I have read the Review entitled (An update on the study of the molecular mechanisms involved 2 in autophagy during bacterial pathogenesis) with a great attention. The review is well organized and updated with most recent references and discusses the whole situation thoroughly. I have no comments to the authors.
>>Response: Thank you very much for taking the time to review our manuscript, "An update on the study of the molecular mechanisms involved in autophagy during bacterial pathogenesis." We appreciate your positive feedback and are pleased to hear that you found our review well-organized and thorough. Your recognition of the comprehensive nature of our work and the use of recent references is encouraging. We are glad that the review met your expectations and provided a clear and updated overview of the field.
Thank you once again for your valuable feedback and for supporting our work.